# Stable Tetrodotoxin Production by *Bacillus* sp. Strain 1839

**DOI:** 10.3390/md17120704

**Published:** 2019-12-13

**Authors:** Daria I. Melnikova, Anna E. Vlasenko, Timur Yu. Magarlamov

**Affiliations:** 1A.V. Zhirmunsky National Scientific Center of Marine Biology, Far Eastern Branch, Russian Academy of Sciences, Vladivostok 690041, Russia; dmelnikova@imb.dvo.ru (D.I.M.); avlasenko@imb.dvo.ru (A.E.V.); 2School of Biomedicine, Far Eastern Federal University, Vladivostok 690090, Russia

**Keywords:** tetrodotoxin, TTX, *Bacillus*, spores, TTX-producing bacteria, HPLC-MS/MS

## Abstract

For the first time, tetrodotoxin (TTX) was detected in a bacterial strain after five years of cultivation in laboratory conditions since its isolation from the animal host. A reliable method suitable for bacterial samples, high-performance liquid chromatography with tandem mass spectrometry, was used for toxin detection in spore and vegetative cultures of *Bacillus* sp. 1839. TTX was detected in a spore culture of the strain.

## 1. Introduction

Tetrodotoxin (TTX), being one of the most famous low molecular weight neurotoxins capable of voltage-gated sodium channels blocking in nerve and muscle tissues was found in a great variety of marine and terrestrial organisms and sea and freshwater sediments [1]. Its origin in organisms and ecosystems is still a controversial issue, but the discovery of numerous bacterial strains capable of TTX production isolated from the above-mentioned sources indicates the link between bacteria and toxin occurrence. The existence of a bacterial strain proved to produce TTX independently of the host organism can dramatically facilitate the discovery of still unknown pathways of TTX biosynthesis. Nevertheless, one of the main concerns associated with TTX-producing bacteria is the inability of most found bacteria to produce toxin through long-time cultivation or even after several passages [2]. Another important question is the method for TTX detection. Despite different methodological approaches based on antigen specificity, neurotoxic effect, or physicochemical properties of the toxin used by researches in studies with bacteria, only liquid chromatography with tandem mass spectrometry could be considered the most specific and reliable to date [1]. 

Our research group worked on the problem of TTX distribution in marine ecosystems with a special emphasize on a screening of bacterial TTX-producers for several years. Search for TTX-producing bacteria in the toxic TTX-bearing ribbon worm *Cephalothrix simula* using confocal laser scanning microscopy with polyclonal anti-TTX antibodies, held in 2014, revealed TTX-positive labeling in bacterial cells of *Bacillus* strain [3]. Detailed investigation of the *Bacillus* sp. strain 1839 using immunoelectron microscopy with anti-TTX antibodies showed a strict association of TTX labeling with endospores and free spores of bacteria [4]. Further investigations of the life cycle [5] and sporulation conditions [6] showed that the strain was TTX-positive even after numerous passages for three years since its discovery, which, in combination with spore-associated TTX synthesis, makes it unique among other TTX-producing bacteria. This indicates the importance of the confirmation of TTX production by *Bacillus* sp. 1839 by more reliable methods of toxin detection.

Current research is the first report of TTX synthesis by bacteria after five years since its isolation. TTX was revealed in the spore culture of *Bacillus* sp. strain 1839 using high-performance liquid chromatography with tandem mass spectrometry (HPLC-MS/MS).

## 2. Results

As a result of the HPLC-MS/MS analysis of spore and vegetative culture of *Bacillus* sp. 1839, TTX was detected (Figure 1A, Table 1). Toxin was found only in the spore culture of the strain. The MS/MS fragmentation spectrum of *Bacillus* sp. 1839 spore culture extract showed characteristic fragment ions of TTX (M + H)^+^ m/z 320 precursor: (M + H-H_2_O)^+^ at m/z 302 and (M + H-C_3_H_7_O_6_)^+^ at m/z 162 (Figure 1B).

In the spore culture of the strain peak with the retention time (Rt) 5.66 min and MRM transitions 272.10 > 254.10 and 272.10 > 162.10, presumably related to 5,6,11-trideoxyTTX, was found. However, its concentration was lower than LoQ (limit of quantification, <0.6 ng/mL), and the MS/MS fragmentation spectrum was not obtained, not allowing to confirm 5,6,11-trideoxyTTX presence in the bacterium.

## 3. Discussion and Conclusions

From the very early discovery of the first TTX-producing symbiotic microflora in the intestine of the crab *Atergatis floridus* [7] till nowadays, representatives of 31 genera of bacteria were shown to contain TTX and its analogs [1]. However, the production of TTX in vitro by any of the discovered bacterial strains was not optimized. Changes in growth conditions, such as temperature and media content, were reported to increase the TTX synthesis. For example, early research of *Alteromonas tetraodonis* noted the phosphate concentration in medium to affect the TTX production [8]. Recent works showed better TTX yield because of cultivation at lower temperatures in a range of 22–25 °C [9,10]. In a number of studies, TTX production depended on the growth phase of the bacteria and time of cultivation, respectively [8,11]. Moreover, the results of these studies were controversial. In the case of *Alteromonas tetraodonis*, TTX production was associated with the stationary growth phase [8]. On the other hand, Yu et al. [11] showed that toxin synthesis in the culture of *Raoultella terrigena* in its log growth phase was almost twice that in the stationary phase. An interesting study conducted by Liu et al. [12] showed a correlation of TTX concentration with the number of copies of the pNe-1 plasmid in cells of *Aeromonas* sp. Ne-1. The authors suggested that the bacterium might have the ability to transfer the TTX biosynthesis gene via the conjugation and contagion of plasmid pNe-1. Despite TTX detection in bacterial cultures during the first hours of cultivation, as in the case with the *Aeromonas* sp. Ne-1, which lost the plasmid with the ability to synthesize TTX after 18 h of culturing, as, in most other TTX-positive strains, bacteria were not able to produce toxin through time in laboratory conditions.

In the case of *Bacillus* sp. 1839, TTX synthesis is directly linked with the sporulation phase of the bacterial life cycle. Immunoelectron microscopy with anti-TTX antibodies, held in 2014, reveled TTX-positive labeling in the cytoplasm of the mother cell, coat, cortex, and core of forespores and integument and the core of free spores of the bacterium [4]. This unique characteristic indicates the possibility of long-time cultivation of the strain leading to an increase in TTX production. Results of the current research confirmed previous works and revealed TTX presence only in the spore-enriched, not vegetative, culture of the strain incubated for seven days. Moreover, extensive use of *Bacillus* sp. 1839 in different experiments and numerous passages did not lead to the loss of TTX-producing ability of the strain. Because of the high specificity of HPLC-MS/MS in TTX detection, we have reliable data for TTX-production by *Bacillus* sp. 1839.

Having a complex structure composed of a guanidinium moiety bound to a highly oxygenated carbon skeleton with a 4-dioxaadamantane portion containing five hydroxyl groups [13], TTX leaves plenty of questions about its biosynthetic pathways. There are several proposed pathways of TTX biosynthesis from arginine involving the incorporation of a guanidinium moiety with amidinotransferase or non-ribosomal peptide synthetase, and the carbon backbone origin through polyketide, C5 branched sugar, or C5 isoprene [13]. An important tool allowing predicting TTX biosynthesis is the structure of its naturally occurring analogs found in a wide variety of TTX-bearing animals [14]. According to Yotsu-Yamashita et al. [15], the late stages of biosynthesis and metabolism of TTX and its analogs may involve two oxidation routes of 5,6,11-trideoxyTTX to TTX: First with the oxidation to 5,11-dideoxyTTX followed by oxidation to both of 5-deoxyTTX and 11-deoxyTTX; second through the conversion to 6,11-dideoxyTTX followed by oxidation to 11-deoxyTTX. As the conversion between non-equilibrium toxin analogs and TTX in animals has never been detected, oxidation reactions are supposed to proceed in TTX-producing microorganisms [15,16,17]. To date, no TTX analogs except chemically equilibrium to toxin (4-epiTTX and anhydro-TTX) were found in TTX-producing microorganisms [1]. In the present study, chromatographic peak presumably related to the 5,6,11-trideoxyTTX was detected in the spore culture of *Bacillus* sp. 1839. Nevertheless, analytical data obtained in current research are not strong enough, and further studies aimed at searching for TTX analogues in bacteria should be conducted.

It is not a secret, that bacteria, having a relatively small genome in combination with widely known mechanisms of gene regulation, are the most preferable model system for the study of different biosynthetic routes. The existence of the bacterial strain with a stable TTX production constant through time gives a perfect possibility to reveal toxin biogenesis. We suggest *Bacillus* sp. 1839 as a good candidate for future researches dedicated to the unraveling of molecular mechanisms of TTX synthesis. Further investigations with the strain will include the study of the genome and transcriptome on different stages of the life cycle of bacterium and works connected with gene expression regulation.

## 4. Materials and Methods 

### 4.1. Bacterial Strain and Growth Conditions

The strain *Bacillus* sp. 1839 (KF444411-KF444416) was obtained from the Collection of Marine Heterotrophic Bacteria, A.V. Zhirmunsky National Scientific Centre of Marine Biology, Far Eastern Branch of the Russian Academy of Sciences. Plates with DifcoTM Marine Agar 2216 (BD, USA) (pH 7.6) were used for bacterial cultivation. To obtain a vegetative cell culture, bacteria were incubated at 23 °C for 2 days. Spores-enriched cell culture was obtained by incubation at 23 °C for 7 days until spores content exceeded 50%. 

The modified Abbott’s method was used to reveal the spores [18]; the smear was stained with methylene blue (Sigma-Aldrich, St. Louis, MO, USA) solution with 1% NaOH and then counterstained with Neutral Red (Sigma-Aldrich, St. Louis, MO, USA). The specimens were examined under an Olympus IX83 light microscope (Japan). Bacteria were harvested by adding a sterile physical solution to each agar plate and scraping off the biomass using an L-shaped spreader, with subsequent centrifugation (14,000× *g*, 20 min, 4 °C) and the supernatant removal. The resulting pellet was stored at −20 °C until further investigation.

### 4.2. Toxins Isolation and Purification

The extracts were prepared using the following procedure: bacterial pellet was homogenized in 1% acetic acid solution 1:10 (vol/vol) for 10 min using homogenizer FastPrep-24™ (MP Biomedicals, USA) (4.5–5.5M, 10 cycles, 60 sec. each); resulting suspension was centrifuged (14,000× *g*, 30 min, 4 °C), and the supernatant was taken; pellet was washed twice with 1% acetic acid solution, centrifuged, and the supernatant was taken out and pooled. The extract clean-up was carried out with SPE Cartridge, Chromafix C18 ec (S) (Macherey-Nagel GmbH & Co., Germany). The extract was aspirated through the column, the column was washed with 1% acetic acid (0.5 vol of sample), then filtrates were pooled. To remove the proteins, the extract was heated for 5 min at 95 °C and centrifuged (14,000× *g*, 20 min, 4 °C), final supernatant was stored at −20 °C until further investigation. 

Toxins purification was performed using an activated charcoal column according to the following protocol: 100 mL of activated charcoal (Sigma-Aldrich, USA) was placed into the Vivaspin turbo centricons with molecular cut-off of 300 kDa (Sartorius, Germany) and equilibrated with water. Total of 5 mL of bacterial extract was put on the column, charcoal was resuspended, mixture was incubated for 5 min at room temperature, and centrifuged at 700× *g* for 5 min. Then column was washed twice with 5 mL of distilled water. To eluate toxins the charcoal in the column was resuspended with 1 mL of 1% acetic acid in 20% ethanol, incubated for 5 min at room temperature, and centrifuged; this step was repeated ten times. Toxins purification procedure was repeated until the entire extract was treated. Eluates were pooled and evaporated to dryness in vacuo. The resulting precipitates were dissolved in a 0.1% aqueous acetic acid solution in ratio of 50 µL per 1 mL of bacterial pellet and analyzed for TTX and its analogs presence by HPLC-MS/MS (HPLC-MS/MS analysis was conducted in School of Biomedicine of Far Eastern Federal University) according to the procedure of Bane et al. [19] with modifications (see below).

### 4.3. Toxins Identification by HPLC-MS/MS

HPLC-MS/MS analysis was conducted using a Shimadzu LC system coupled to a triple quadrupole mass spectrometer Shimadzu 8060 (Kyoto, Japan). Separation was carried out on a SeQuant ZIC HILIC column, 5 μm, 150 × 2.1 mm (Merck) at 40 °C, at a flow rate of 200 μL/min. A binary gradient was deployed, composed of mobile phase A, ammonia (5 mM), and formic acid (8 mM) in 94:6 acetonitrile: water and mobile phase B, ammonia (10 mM) and formic acid (20 mM) in water. A gradient profile was used as follows: (a) 0–4.3 min 15% B; (b) 4.3–16 min 25% B; (c) 16–20 min 50% B. A guard column SeQuant ZIC HILIC (20 × 2.1 mm, 5 μm) (Merck, Germany) was installed in line before the analytical column through a 2-position 6-port valve. At 4.4 min valve was switched and the guard column backflushed with isopropanol (4.4–9 min) and water (9–15 min) at flow rate 0.3 mL/min. At 16 min valve was switched back. The mass spectrometer was operated in the scan mode (m/z 200–1000) and multiple reaction monitoring (MRM) mode. The ion source parameters are as follows: interface temperature–380 °C, desolvatation line temperature–250 °C, nebulizing gas (N_2_) flow–3l/min, drying gas (N_2_) flow–3l/min, heating gas (dry air) flow–17 l/min. To detect toxins, MRM transitions of TTX analogs from Bane et al. [14] were used. MRM transitions for detected toxins were: 272.10 > 254.10 (collision energy (CE) was 41 eV) and 272.10 > 162.10 (CE = 25 eV, Rt 5.66 min) for 5,6,11-tridoxyTTX; 320.10 > 302.10 (CE = 41 eV) and 320.10 > 162.10 (CE = 25 eV, Rt 14.63–14.66 min) for TTX. MS/MS was performed in positive ion mode, CE was 35 eV. A mixture of TTX analogs containing 5,6,11-trideoxyTTX prepared from the extract of nemertean *C. simula* according to Vlasenko et al. [20] was used as a standard. TTX concentration was calculated using calibration curve of standard TTX solution series (Alomone Labs Ltd., Jerusalem, Israel). Calculation of TTX analogs concentration was carried out according to the procedure proposed by Chen et al. [21]. The toxins detection criteria were the ratio S/N > 3 of the precursor MRM transition peak, the relative intensity of the fragment ion peak > 4%, the order of toxins elution according to Bane et al. [19]. The method was validated using standard TTX solutions in MRM mode. The linearity range was from 0.6 to 100 ng/mL, the recovery range from 1 to 100 ng/mL of TTX was 98.4%, the LoQ was 0.6 ng/mL, the limit of detection (LoD) is 0.2 ng/mL, and the relative standard deviation was 4.5–14.6%. No recovery correction was made.

## Figures and Tables

**Figure 1 marinedrugs-17-00704-f001:**
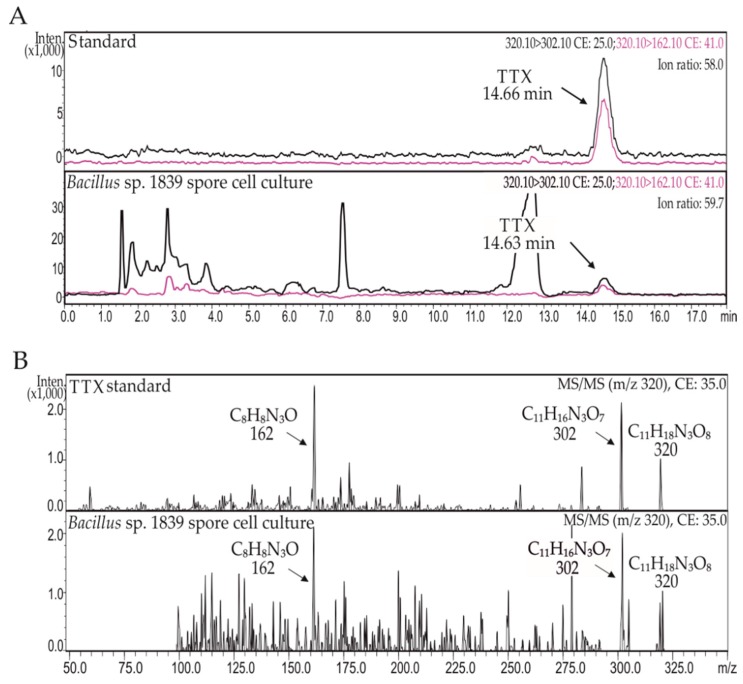
(**A**) High-performance liquid chromatography with tandem mass spectrometry (HPLC-MS/MS) chromatograms of tetrodotoxin (TTX) standard and *Bacillus* sp. 1839 spore culture extract; (**B**) MS/MS spectra of TTX standard and *Bacillus* sp. 1839 spore culture extract. As a standard for TTX, a commercial TTX solution was used (C_TTX_ = 1 ng/mL).

**Table 1 marinedrugs-17-00704-t001:** TTX concentration in *Bacillus* sp. 1839 vegetative and spore cell cultures.

*Bacillus* sp. 1839 Culture	Toxin Amount	TTX
Vegetative cell culture	ng/mL of extract	-
	ng/L of pellet	-
Spore cell culture	ng/mL of extract	0.751
	ng/L of pellet	30.04

-: not detected.

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
