# Peer review of "Stable Tetrodotoxin Production by Bacillus sp. Strain 1839"

_marinedrugs, 2019, doi:10.3390/md17120704_

Round 1
Reviewer 1 Report
I was excited to see this manuscript, and delighted that work is being conducted to hopefully show evidence for reliable, repeatable production of TTX (and even associated analogues) from marine bacteria. The paper is well written in many ways, with a good background and the aims are clear.
However, I became dissapointed when I saw the results as they are currently presented. Quite simply, I do not believe this is enough evidence to conclusively say TTX and analogues are present. The chromaotgraphic interferences are enormous, so picking out a peak in amongst the large number of other peaks without providing additional supporting information is inadequate.
TTX itself is not as bad, but I woudl still rather see data showing relative ion ratios between std and sample for this analyte. For the other analytes, I just dont think the evidence as currently presented is enough.
For this to be accepted, I think more evidence is needed, including ion ratios, mass spectra. Ideally, I would like the analysis to be conducted after some sort of clean-up or concentration, to provide a strong signal, thereby making the data more clear.
As it stands, claiming reliable production of TTXs from culture is not justifiable, but this can be improved if the authors take these steps to rectify the presentation
Author Response
We thank this reviewer sincerely for constructive criticism and valuable comments that have helped improve our manuscript. We also thank this reviewer for the overall positive feedback for our paper. Our specific response is provided below.
“However, I became dissapointed when I saw the results as they are currently presented. Quite simply, I do not believe this is enough evidence to conclusively say TTX and analogues are present. The chromaotgraphic interferences are enormous, so picking out a peak in amongst the large number of other peaks without providing additional supporting information is inadequate. TTX itself is not as bad, but I would still rather see data showing relative ion ratios between std and sample for this analyte. For the other analytes, I just dont think the evidence as currently presented is enough.
For this to be accepted, I think more evidence is needed, including ion ratios, mass spectra. Ideally, I would like the analysis to be conducted after some sort of clean-up or concentration, to provide a strong signal, thereby making the data more clear”.
Response:
We agree with this reviewer that data about relative ion ratios between std and sample is necessary in this case, so we added it to Figure 1 for both TTX and 5,6,11-trideoxyTTX. We also conducted additional analysis and obtained mass spectra for TTX, which was added to Figure 2. Due to the small concentration of 5,6,11-trideoxyTTX in the sample, we are not able to obtain suitable mass spectra, but since MRM transitions are obvious and relative difference of ion ratios for standard and sample is 0,7 % we are sure in this analog and insist on the reliability of this data. Since additional analysis allowed us to obtain better images, we changed chromatograms to Figure 1. What about 11-norTTX-6(S)-ol we agree with the reviewer that chromatogram for this analog is not clear, and despite ion ratios are reliable, we are not able to obtain a better image due to its low concentration. It was excluded from the manuscript.
We also want to pay attention to the difficulties with the analysis of bacterial extracts, which are a low concentration of the desired substances and a large number of impurities. Additional clean-up procedures always lead to the loss of already small quantities of TTX, so in current research, we concentrated on the detection of toxin presence rather than the purity of the extract.
All changed and added text in the manuscript is highlighted with a light grey color.
Reviewer 2 Report
This is an interesting paper that should be accepted. Together prior work from the authors (ref. 6), the present contribution opens the door to the production of TTX by fermentation and to the exploration of its biosynthesis in a convenient laboratory setting.
The paper requires careful proofreading to eliminate typos, garbled sentences, and other problems. Throughout the manuscript, the symbol ml (milliliter) should be changed to mL. The following issues are of special concern:
Line:
72: “toxicity was higher at a log or stationary phase”-- unclear: please rephrase
76-78: “despite positive …laboratory conditions.” – unclear: please rephrase
79-80: of a bacterial --> of the bacterial
102-104: “So far …microorganisms” – unclear: please rephrase
140 For proteins removal from the extract it was --> To remove proteins, the extract was
143: 100 ml of an activated charcoal --> 100 mL of activated charcoal
157: ammonium (5 mM) --> ammonium [ acetate? ] (5mM)
159: 50% B; A guard column --> 50% B. A guard column [ punctuation ]
160: before analytical column --> before the analytical column
161: through 2-position --> through a 2-position
161: and guard columns --> and the guard column [ line 159 indicates that only 1 guard column was used: please verify ]
161: backflashed --> backflushed
163: operated in scan --> operated in the scan mode [ ? ]
172: in Vlasenko --> according to Vlasenko
Author Response
We thank this reviewer sincerely for careful reading of our manuscript and this positive feedback. Our responses are provided below. All changed and added text in the manuscript is highlighted with a light grey color.
“The paper requires careful proofreading to eliminate typos, garbled sentences, and other problems. Throughout the manuscript, the symbol ml (milliliter) should be changed to mL”
Response: We thank this reviewer for valuable comments that helped improve our manuscript. The manuscript was carefully read and checked for grammatical and lexical mistakes. The symbol ml was replaced to mL.
“The following issues are of special concern:
“72: toxicity was higher at a log or stationary phase”-- unclear: please rephrase”
Response: We have changed this phrase as follows: “In the case of Alteromonas tetraodonis, TTX production was associated with the stationary growth phase [9]. On the other hand, Yu et al. [12] showed that toxin synthesis in the culture of Raoultella terrigena in its log growth phase was almost twice that in the stationary phase”. Line 74-77.
“76-78: “despite positive …laboratory conditions.” – unclear: please rephrase”
Response: We have changed this phrase as follows: “Despite TTX detection in bacterial cultures during the first hours of cultivation, as in the case with the Aeromonas sp. Ne-1, which lost the plasmid with the ability to synthesize TTX after 18 h of culturing, as, in most other TTX-positive strains, bacteria were not able to produce toxin through time in laboratory conditions”. Line 80-84.
“79-80: of a bacterial --> of the bacterial”
Response: Done. Line 85-86.
“102-104: “So far …microorganisms” – unclear: please rephrase”
Response: We have changed this phrase as follows: “To date, no TTX analogs except chemically equilibrium to toxin (4-epiTTX and anhydro-TTX) were found in TTX-producing microorganisms [1]”. Line 107-109.
“140 For proteins removal from the extract it was --> To remove proteins, the extract was”
Response: Done. Line 144.
“143: 100 ml of an activated charcoal --> 100 mL of activated charcoal”
Response: Done. Line 147.
“157: ammonium (5 mM) --> ammonium [acetate?] (5mM)”
Response: Done. Line 160-161.
“159: 50% B; A guard column --> 50% B. A guard column [punctuation]”
Response: Done. Line 163.
“160: before analytical column --> before the analytical column”
Response: Done. Line 164.
“161: through 2-position --> through a 2-position”
Response: Done. Line 164.